# Separation and Characterization of Cellulose Fibers from Cannabis Bast Using Foamed Nickel by Cathodic Electro-Fenton Oxidation Strategy

**DOI:** 10.3390/polym14030380

**Published:** 2022-01-19

**Authors:** Ying Sun, Duanxin Li, Yang Yu, Jialin Chen, Wanyue Fan

**Affiliations:** 1College of Light Industry and Textile, Qiqihar University, Qiqihar 161000, China; ldx0711@sina.com (D.L.); annie802316@163.com (Y.Y.); lin177227@sina.com (J.C.); fanwanyue@163.com (W.F.); 2Engineering Research Center of Flax Processing Technology, Ministry of Education, Qiqihar University, Qiqihar 161006, China

**Keywords:** cannabis fibers, electro-Fenton (EF), nickel-foam (Ni-F), oxidation degumming, fiber properties

## Abstract

Degumming is the most important link in the textile industry. The main purpose of degumming is to effectively remove non-cellulose substances in plant bast fibers. In this research, we propose an electro-Fenton (EF) system with a nickel-foam (Ni-F) cathode in weak acid pH (EF/Ni-F) to degum cannabis fiber in EF while reducing the content of pollutants in degumming wastewater. FT-IR, XPS, XRD, SEM, and TG were employed to thoroughly understand the reaction characteristics to characterize chemical components, element qualities, the crystallinity, and the morphologies of degummed fibers. Additionally, physical and mechanical properties such as breaking strength, elongation at breaking, residual glue rate, whiteness, and diameter of degummed fibers were measured. Through testing, it was found that the fiber degummed by the EF method had higher breaking strength, lower residual tackiness, and higher whiteness than other methods. The antibacterial test was used to detect the effect of fiber on Staphylococcus aureus before and after degumming. EF could remove more colloidal components from cannabis than other methods, and the mechanical properties were also enhanced. The characteristics of the degummed fiber further confirmed the effectiveness of the new degumming method. Moreover, the antibacterial experiment found that the antibacterial property of the degummed fiber was enhanced. The colloidal components in the degumming wastewater were flocculated and precipitated. The upper liquid of the solution had low chromaticity, low COD value, and weak acid pH value, which can meet the discharge requirements. The above test proves that EF is an effective degumming method that is environmentally friendly, takes less time, and enhances antibacterial performance.

## 1. Introduction

Cannabis fiber, as a natural and renewable textile fiber [1], has attracted great interest due to its excellent properties in various fields. Examples of these properties are good moisture absorption and breathability [2], antibacterial properties [3], high tensile strength [4], radiation resistance, and biodegradability. The main reason why the earliest fiber plant failed to become one of the important textile crops was its complicated degumming process [5], which simplifies the cannabis fiber because the degumming process is the most important point in development [6]. The current problems in the extraction of cannabis fiber are mainly practical factors such as long extraction process time, high acid-base concentration, large dosage, and environmental pollution by degumming waste liquid. The oxidative degumming method is a rapidly developing method in the field of fiber extraction [7].

The Fenton process is considered the most effective method in advanced oxidation technology. It uses highly active free radicals to oxidatively degrade viscose in a special environment quality and organic matter in water, so that treated wastewater does not produce secondary pollution [8]. Wastewater derived from traditional textile industry degumming has rich color and complex composition, hence, treating wastewater satisfactorily is difficult [9].

In recent years, electrochemical technology has aroused great interest in fiber degumming and subsequent wastewater treatment. This technology can successfully remove the glue in the fiber and various pollutants in industrial wastewater. In the electrochemical oxidation process, the oxide is removed by direct or indirect oxidation process. In the direct oxidation process, organic matter adsorbed on the anode surface is decomposed and transferred by electrons [10]. This process does not need to participate in the oxidizing agent. In the indirect oxidation process, organic matter is decomposed in the solution through an electrochemical oxidation reaction to produce an oxidant, such as hypochlorite, chlorine, and hydrogen peroxide (H_2_O_2_) [11]. Nickel foam (Ni-F) was a new type of electrode with excellent physical and mechanical properties, which was expected to be used in large-scale applications [12].

Electro-Fenton (EF) technology represents the combination of electrochemical process and Fenton oxidation process. In the EF process, the major mechanisms involved in the pollutant degradation are Fenton reaction in the solution (Equation (1)) and direct oxidation on the anode surface (Equation (2)) [13]. By reducing oxygen in the cathode under acidic conditions, H_2_O_2_ is continuously generated in the solution during the electrolysis process (Equation (2)). The electrochemically generated H_2_O_2_ and ferrous ions (Fe^2+^) added to the solution generate hydroxyl radicals (•OH) through the classic Fenton reaction (Equation (1)) [14]. •OH is a powerful oxidant with a high oxidation potential, and it was combined with organic pollutants to ensure complete mineralization. The •OH radical reacts with the glue on the fiber to oxidize and degrade to produce water and small molecular glue dissolved in the solution. As the reaction proceeds, CO_2_ and HO_2_ are produced (Equation (3)) [15]. Using Ni-F as the cathode can further increase the regeneration rate of Fe^2 +^ through Ni reduction (Equation (4)) [16]. The reaction to generate H_2_O_2_ reduces the feeding cost and solves the shortcomings of rapid consumption of H_2_O_2_ as the reaction proceeds. In this process, Fe^2+^ ions are reduced by electroreduction of Fe^3+^ on the cathode (Equation (4)) to reduce the production of iron sludge.

The electric-Fenton system is more often used in the treatment of wastewater such as water-oil separation and difficult-to-treat dyestuffs, with very good results. The composition of pollutants in wastewater is complex, with many types, high concentration of pollutants, high toxicity, high salt content, and difficulty to be biodegraded. It has the characteristics of large discharge, wide pollution, and poor biodegradability, which makes conventional physical, chemical, and biological methods difficult. Meet the technical and economic requirements of purification treatment. If these substances are discharged into the environment without treatment, it is bound to seriously pollute the ecological environment and threaten human health. The principle of the electro-Fenton reagent oxidation process is to add oxidant H_2_O_2_ and catalyst Fe^2+^ to the wastewater in the electrified state. H_2_O_2_ decomposes under the catalytic action of Fe^2+^ and electrified electrodes to produce hydroxyl radicals (•OH) with high reactivity. Through electron transfer and other means, the colloidal organic matter in the wastewater is oxidized and decomposed into small molecules, CO_2_ and H_2_O_2_, so as to achieve the purpose of degrading COD.

At present, there is no research on an electro-Fenton oxidation system for degumming and separating fibers, but few scholars have tried to use Fenton reagent to degum fibers. Zhou [17] performed oxidative degumming of ramie fiber under the Fenton system and found that the degumming effect of ramie is good and the organic content in the degumming waste liquid is low. The COD value of the wastewater degumming by Fenton method is 16,800 mg·L^−^^1^ lower than the traditional degumming COD value of 22,477 mg·L^−1^. At the same time, the pH value of 11.88 is lower than the traditional degumming pH value of 13.11, which also verifies the advantages of Fenton oxidative degumming from the side. Song [18] used the steam explosion method and the Fenton method to extract the kenaf fiber. The experiment found that the combined degumming method did not change the crystal structure of cellulose, but it could gradually degrade hemicellulose, lignin, and other colloidal substances in the amorphous area. Disordered areas in cellulose are used to achieve the purpose of degumming.

Therefore, the use of the electricity and Fenton system combined degumming cannabis fiber as a new attempt. The use of foamed nickel as the cathode accelerates the occurrence of electrical reactions to assist the Fenton system to decompose oxidative free radicals and H_2_O_2_ acts on the external colloid of cannabis fiber, so that the colloid is degraded into small molecules and dispersed in the solution to achieve the purpose of degumming. The electro-Fenton process is not only regarded as an efficient and environmentally friendly method to replace traditional degumming processes for fiber degumming, but also as a solution to the problem of difficult treatment of organic matter in wastewater after degumming.
H_2_O_2_ + Fe^2+^ →•OH + Fe^3+^ + OH^−^(1)
O_2_ + 2H^+^ + 2e^−^ → H_2_O_2_(2)
•OH + Inoglia → •Inoglia + H_2_O→······→ CO_2_ + HO_2_(3)
2Fe^3+^ + Ni → 2Fe^2+^ + Ni^2+^(4)

## 2. Experimental

### 2.1. Material

Cannabis material originated from Heilongjiang Province. Nickel-foam (Ni-F) was sourced from Jiangsu Guangshengjia New Material Co., Ltd., Jiangsu, China. The chemical composition in cannabis fiber was analyzed in accordance with China National Standard GB 5889-86 (method of quantitative analysis of ramie chemical components) listed in Table 1.

### 2.2. Chemicals

The main chemicals used in this study were iron sulfate heptahydrate, hydrogen peroxide (30%, *w/w*), sulfuric acid, sodium hydroxide, sodium tripolyphosphate, and sodium silicate, which were purchased from Sinopharm Chemical Reagent Co., Ltd., Shanghai, China. The pH of the solution was adjusted to desired value (6.0) by using the (98%, *w/w*) sulfuric acid. All chemicals used in this study were analytical grade reagents without any further treatment, and all solutions were prepared with distilled water.

### 2.3. Degumming with Biological Enzyme

The weight of the cannabis sample used for each degumming process was around 2.0 g and the dosages of chemicals were the weight percentage on cannabis fibers (*w/w*). The total amount of biological enzymes was 10% of the fiber weight. Table 2 shows the conditions of biological enzyme degumming [19].

### 2.4. Oxidation Degumming in Alkaline Condition

The dosages of chemicals were the weight percentage on cannabis fibers (*w/w*). The oxidation degumming in alkaline condition were shown in Table 3 [20].

### 2.5. Oxidative Degumming with Electro-Fenton Reagent

The dosages of chemicals were the weight percentage on cannabis fibers (*w/w*). The oxidative degumming with electro-Fenton reagent treatment conditions were shown in Table 4.

### 2.6. Constituent Content Test

The chemical composition of the cannabis fiber was tested. The process included the fat wax, water-soluble substances, pectin, hemicellulose, ash, lignin degradation, and cellulose extraction in the cannabis fiber.

### 2.7. Residual Glue Rate Test

The residual glue rate of cannabis was analyzed according to the China National Standard GB 5889-86 (method of quantitative analysis of ramie chemical components).

The residual glue rate was calculated with Equation (5) [21]:(5)$$\mathrm{The}\;\mathrm{residual}\;\mathrm{glue}\;\mathrm{rate}=\frac{G_{0}-G_{1}}{G_{0}}$$
where $G_{0}$ is the dry weight of the fiber after degumming, and $G_{1}$ is the dry weight of the hemp fiber after treatment.

### 2.8. Mechanical and Physical Test

Fiber samples were conditioned in standard atmospheric condition (temperature 20 ± 2 °C, RH 65 ± 2%) for 24 h prior to test. Tenacity and elongation at breaking were tested using an LLY-06E fiber strength instrument under 20 °C and RH 65% condition. The pre-tension was 0.4 cN/dtex, and the gauge length and drawing speed were kept at 10 mm and 30 mm/min, respectively. Average values were obtained using results from 20 specimens. The diameter of the fiber was measured 20 times for each sample by the YGB-002 fiber fineness meter, and the average value was used as the final result. The whiteness of the fiber was measured by a YQ-Z-48A whiteness measuring machine for each sample at different locations for 10 times, and the average value was used as the final result.

### 2.9. FT-IR Analysis

FT-IR analysis was employed to determine the chemical functional groups in treated fiber. The fiber was analyzed using Spectrum One infrared spectrum analyzer (PE, USA). The spectra obtained were the results of 30 scans within the range of 400–4000 cm^−1^ at a resolution of 4 cm^−1^.

### 2.10. XPS Analysis

XPS analysis of the fiber surface was taken using ESCALAB250Xi (Thermo, Waltham, MA, USA) photoelectron spectrometer to test the chemical composition of the sample surface.

### 2.11. XRD Analysis

X-ray diffraction (XRD) patterns were recorded from 2θ = 7–60° with a D/max-RB diffractometer equipped with a graphite monochromator and Cu Kα radiation at λ = 0.154 nm (45 kV, 200 mA).

### 2.12. SEM Analysis

Surface micrographs of fiber were tested by an S-3400 (Hitachi, Tokyo, Japan) scanning electron microscope. It was operating at 10 kV, temperature 20 °C, and RH 65%. Prior to SEM evaluation, the samples were coated with a thin layer of gold by means of a plasma sputtering apparatus.

### 2.13. TGA Analysis

Dynamic thermogravimetric measurements were performed by using a STA449F3 instrument (Netzsch, Selb, Germany). Temperature programs for dynamic tests were run from 25 °C to 600 °C at a heating rate of 10 °C/min. These tests were carried out under nitrogen atmosphere (20 mL/min) in order to prevent any thermoxidative degradation.

### 2.14. Antibacterial Test

The antibacterial ring test method was used in this study to evaluate the antibacterial properties of fibers. Nutrient agar is used as the growth medium. Staphylococcus aureus (ATCC^®^ 6538™) was considered as the representative bacteria to evaluate the response of the test sample.

### 2.15. Degumming Waste Liquid Test

The COD of degumming wastewater was assayed after the international standard JJG 1012–2006 [21]. The pH value was tested by a precision acidity meter (Hangzhou Qiwei Instrument Co., Ltd., Hangzhou, China).

## 3. Results and Discussion

### 3.1. Chemical Composition of Different Degumming Methods of Cannabis Fiber

Lignin plays an important role in the structure of all woody plants, as it helps hold fiber bundles together [22]. A high percentage of lignin may result in a harsh handle of cannabis fiber [23]. Hemicellulose is susceptible to alkali attack, and cellulose has strong resistance to alkali attack [24]. A low fiber residual glue rate after treatment indicates to some extent that the removal rate of hemicellulose, pectin, and lignin is high, thus resulting in fine fibers.

Usually, the residual glue rate is used to prove the degumming effect, which can directly reflect the degree of gum removal. However, a certain amount of residual gum will improve the performance of degummed fiber and the subsequent spinning performance of the yarn [25]. Therefore, it is as far as possible to remove the non-cellulose component of the gum outside the fiber, but due to the thick and hard characteristics of the cannabis fiber, in order to make the subsequent spinning process smoothly still retain a part of the gum content.

Table 5 shows the chemical composition of cannabis fiber under different degumming methods. It shows that irrespective of the treatment method, the cellulose content of the degummed fiber increased compared with the untreated raw cannabis. The biological enzyme in the degumming method takes a long time to take effect, but the removal of colloidal materials is not obvious. The lignin content decreased from 18.98% to 9.43%, and the hemicellulose content decreased from 19.32% to 6.84%. Therefore, the biological enzyme method can effectively remove lignin and hemicellulose. However, oxidative degumming under alkaline conditions has a good effect on the removal of lignin content and other components, but the effect of reducing hemicellulose content is not significant. Enzyme treatment can specifically and efficiently remove hemicellulose and lignin in fiber. However, under peroxide conditions, lignin is converted into oxidized lignin, and it takes a certain time and concentration to dissolve in alkaline solution due to the oxidation reaction destroying the three-dimensional structure between lignin monomers and weakening the intermolecular forces. The role of alkali in degumming is to promote the decomposition of hydrogen peroxide, destroy the structure of hemicellulose, and reduce the fiber gum content through multiple effects. Compared with the oxidative degumming under alkaline conditions, treatment with EF reagent can further reduce gum material, whereas the cellulose content further increases it, and the residual gum rate is significantly reduced. Therefore, EF can obtain more separated cellulose fibers than the previous method.

### 3.2. Mechanical and Physical Test

The mechanical and physical properties of cannabis fibers are important for industrial applications, and such properties can facilitate the use of cannabis fibers in many applications [26]. The mechanical properties of degummed fiber have greatly improved compared with untreated raw cannabis. Ideally, fiber properties have great toughness and elongation; hence, the best degumming operation parameters are selected on the basis of these properties.

Fiber surface treatment has great influence on the breaking performance of cannabis fiber [27]. Table 6 shows the mechanical and physical properties of fibers under different degumming methods. It shows that biological enzymes, alkaline oxygen treatment, and EF oxidation cause a certain degree of damage to the toughness of cannabis; the elongation becomes longer, the fiber diameter becomes smaller, and whiteness improves. These effects were the result of fiber microstructure cracking and fiber damage caused by the strong extraction reaction [28]. Large molecules were split into small molecules, and small molecules were dissolved in water and removed by multiple washings. The adhesive components were completely removed from the cannabis. The fiber after oxidative degumming with EF had a higher purity of cellulose content, which helps improve the mechanical properties of fiber [29].

### 3.3. FT-IR Analysis

Figure 1 shows the FT-IR spectra of the raw cannabis and degummed fibers processed by different methods. FT-IR analysis was performed to further verify the chemical composition changes.

These peaks were sharp at 670 cm^−1^ (C-OH out-of-plane bending) and 1160 cm^−1^ (C-O-C asymmetric stretching mode), and the peak wind strength decreased after degumming. The value 1210 cm^−1^ corresponds to the peak bending vibration of the cellulose -OH surface after degumming, indicating that the cellulose content increases regardless of the degumming method. The band intensity near 1650 cm^−1^, corresponding to carboxyl-containing uronic acid, is a characteristic peak of pectin. With different degumming methods, the relative strength peak under this characteristic decreases, indicating that pectin content is reduced. This observation suggests that pectin is almost completely removed. A significant absorption difference of 2850 cm^−1^ (-CH_2_ symmetric stretching vibration) is evident, and the absorption peak of the treated fiber almost completely disappeared compared with the untreated fiber that used different degumming treatment methods. The range of 3000 cm^−1^ to 3600 cm^−1^ corresponds to -OH tensile vibration, which is mainly attributed to the structure of cellulose and lignin [30]. According to different degumming methods, the relative strength decreases at 3000–3600 cm^−1^, which indicates that the hydroxyl groups are significantly reduced and removed. The hydrogen bond structure in cellulose is partially destroyed, which leads to mechanical loss in the performance of the degummed fiber.

Figure 1 shows the IR spectrum. Moreover, spectral analysis reveals that most of the non-cellulose materials have been successfully removed from cannabis under EF reagent degumming.

### 3.4. XPS Analysis

Figure 2 shows the XPS spectrum, C1s spectrum, O1s spectrum, and N1s spectrum of raw cannabis and degummed fiber. The main elements detected from cannabis fiber were C, O, N, and Fe. Table 7 shows the binding energy and atomic content of raw cannabis and degummed fiber. It shows that the C1s content of raw cannabis was higher than that in biological enzyme samples, and the O1s content was lower than that in biological enzyme samples. In addition, raw cannabis contains N1s and Fe2p, which disappear in biological enzyme samples. The C1s content of the alkaline–oxygen degumming sample and the EF degumming sample is higher than that of the raw cannabis, and the O1s content was lower. N1s decreased in alkaline oxygen degumming but increased in EF samples. The electron donor is coordinated with Fe^3+^ ions to form a complex; hence, Fe^3+^ ions are bonded through the nitrogen and oxygen atoms in the cannabis fiber [31]. Moreover, excess Fe^3+^ and Fe^2+^ ions are consumed as the subsequent reaction proceeds to form •OH radicals and •HO_2_ radicals with oxidizing properties [32]. These oxidizing groups react with the fiber to remove the gum in it.

### 3.5. XRD Analysis

The XRD pattern (Figure 3) comparisons of raw cannabis and degummed fibers. It shows that these four curves exhibit main crystal peaks of 2θ between 22° and 23°, which corresponds to the (002) crystal plane family of cellulose I. The other peaks of 2θ appear between 14.6° and 17.2°, corresponding to the (101) crystal plane family of cellulose II [33]. Obviously, in biological enzymatic degumming, oxidative degumming in an alkaline environment and EF degumming methods, crystal morphology does not change after degumming. Hence, the crystal morphology remains unchanged during any degumming process. Owing to the removal of amorphous non-cellulose compounds, the absorption peaks of the fibers after biological enzyme treatment reflect greater strength than raw cannabis. The test results of oxidative degumming under alkaline conditions or using EF prove the same growth trend.

By comparing the crystallinity of raw cannabis fiber and degummed fiber XRD spectra, the crystallinity of raw cannabis is 16.3%, the crystallinity of the fiber after biological enzymatic degumming is 53.42%, the crystallinity of alkaline oxygen degummed fiber is 57.39%, and the crystallinity of EF degummed fiber is 55.86%. Obviously, after degumming, the proportion of crystal areas increases, and the crystallinity of cannabis fiber is significantly improved.

This finding indicates that the degummed fiber using EF has relatively high crystalline cellulose and may show a better hand feeling.

### 3.6. SEM Analysis

Figure 4 shows the SEM micrographs of raw cannabis and degummed fibers by different methods at the same magnification (1.0 k× and 2.0 k×). As shown in Figure 4, the internal structures of raw cannabis fibers are closely related, connected by rich colloidal substances. The fiber surface is rough and irregular. Biological enzyme treatment resulted in obvious separation of cannabis fibers, and fiber bundles with less gummy substances appeared. After alkaline oxygen treatment, most of the adhesive material is removed from the fiber surface, and the surface becomes smooth and clean. However, the diameter of the fiber becomes thicker because the entry of the lye causes the fiber to swell. Nonetheless, the gelatinous substance on the surface of cannabis can still be seen. EF-treated fibers were observed, and the morphology of single fibers was observed. The results can prove that the EF treatment is effective for the separation and degumming of cannabis fiber. This finding is also supported by the previous studies [34].

### 3.7. TGA Analysis

Thermogravimetric analysis (TGA) was performed on the cannabis fiber samples treated with original, biological enzyme, alkaline oxygen treatment, and EF treatment to compare their degradation characteristics at different stages of preparation. Figure 5 shows the thermogravimetric and derivative thermogravimetric curves of raw cannabis and degummed fiber. Mass loss was mainly divided into three stages. The weight loss in the first stage ranged from 50–120 °C, which was mainly due to vaporization and removal of bound water in the cellulose sample [35]. The temperature range of 220 °C to 300 °C was due to the thermal depolymerization of cellulose and the cleavage of glycosidic bonds of cellulose [36]. The broad peak of the second stage of weight loss in the range of 230 °C to 500 °C was caused by the lignin component, and the degradation of cellulose occurs between 275 °C to 400 °C [37]. Considering that the thermal stability of hemicellulose was lower than that of lignin and cellulose, with the degradation of hemicellulose, weight loss suddenly decreases at 250 °C. The degradation of the third stage of weight loss at 375 °C was due to the thermal decomposition of cellulose. It can be seen that the final residual amount of the four fiber samples is 10–20%. Because the components in the fiber changed the cracking method under the high heat state, the coke was finally formed and turned into a residue.

The thermal analysis shows significant changes from raw cannabis to EF-treated fibers. The peak DTG temperature of raw cannabis at 345.35 °C, and the peak temperature of DTG of degummed fiber at 356.52 °C, 358.33 °C, and 363.17 °C, indicating that the degummed fiber has a higher degradation temperature than the original fiber. In the raw cannabis fiber, cellulose was surrounded by lignin, hemicellulose, and pectin, which may receive heat earlier from the outside to be charred and accelerate the onset of thermal degradation [38].

### 3.8. Antibacterial Test

The antibacterial activity of raw cannabis and fibers degummed by using different methods against Staphylococcus aureus bacteria at different times (24 h, 36 h, 72 h) was compared. Figure 6 and Table 8 show antibacterial graphs and antibacterial bacterial inhibition zones of raw cannabis and degummed fiber. As shown in Figure 6 and Table 8, the raw cannabis fiber showed inhibited bacterial growth but showed a zone of inhibition of 9.4 ± 0.3 mm. When the biological enzymes degummed the cannabis fiber, as shown in Figure 6, the inhibition zone of 10.2 ± 0.8 mm was displayed. Interestingly, when the cannabis is degummed by alkaline oxygen treatment, the inhibitory effect of bacterial growth becomes smaller, reaching 9.2 ± 0.6 mm. When EF is used to degum cannabis fiber, the inhibition zone reaches 10.4 ± 0.5 mm, effectively inhibiting bacterial growth. Regardless of the degumming method, as time increases, the zone of inhibition becomes smaller.

The antibacterial ability of degummed fiber is generally worse than that of raw cannabis because cannabis contains bactericidal and anti-inflammatory ingredients such as cannabinol. However, the antibacterial performance of biological enzyme treatment is better than that of raw cannabis because biological enzymes have greater antibacterial activity against bacteria. This result reflects obvious antibacterial ability [39]. In the EF experiment, the cannabis fiber after degumming treatment even has a larger inhibition zone than that of raw cannabis fiber. This may be because the reagent used for degumming treatment contains Fe^2+^, which is not completely cleaned and some amount remains in the fiber [40], which is near neutral. Dissolved Fe^2+^ releases •OH in a high-temperature environment [41], causing it to enter the bacteria to undergo Fenton reaction. In addition, Fe^2+^ has antibacterial properties and makes the fiber after degumming treatment more excellent. Thus, the EF method for improving the antibacterial properties of cannabis fiber needs further research.

### 3.9. Degumming Waste Liquid Test

Figure 7 and Figure 8 are, respectively, the top view and the front view of the solution of different degumming methods. In order to study whether the degumming in electro-Fenton has a worse impact on the environment than the commonly used degumming system, the COD value of wastewater was tested. Figure 9 shows the pH value and COD value in the degumming waste liquid. It can be seen from the Figure 7c and Figure 8c that electro-Fenton oxidative degumming has a lighter color than biological enzyme degumming (a) and alkaline oxygen degumming (b), and organic matter such as colloids in EF flocculates into iron sludge to facilitate subsequent processing. 

It can be seen from Figure 9 that the COD value of the EF system is much lower than that of the alkaline oxygen degumming system and the biological enzyme degumming system, so the colloidal organic matter degummed is dissolved in the wastewater and flocculates and settles well. The COD value of EF wastewater is 4985 mg/L, which is lower than the alkaline oxygen degumming (about 24,000 mg/L) and the biological enzyme degumming system (about 18,000 mg/L). The pH value of the degumming waste liquid in the electro-Fenton step is neutralized to neutrality by the subsequent alkaline degumming waste liquid from weak acidity, and the COD value is also reduced, which is more in line with environmental protection requirements.

## 4. Conclusions

This article introduced a new method that used foamed nickel as the cathode in the EF system, a new type of oxidative degummed cannabis fiber under weak acid conditions. Through this method, non-cellulose components were effectively removed from raw cannabis fiber, cellulose components were retained, and the residual gum rate was reduced to the lowest degree. Compared with the biological enzyme degumming method and the alkaline oxygen degumming method, the strength and elongation of the fiber prepared by using the proposed method were enhanced. Although the diameter was reduced, the whiteness was slightly worse than that of other methods. FTIR further verified the removal of hemicellulose and lignin. XPS and XRD found that with the removal of colloidal substances, the crystallinity of the fiber increased. SEM verification showed that the fiber surface was smooth, and the TGA clearly showed that different components lost weight before and after degumming. The antibacterial experiment explored the improvement of antibacterial performance of fiber after using EF degumming. The pH value in the degumming waste liquid was close to neutral, the chromaticity was reduced, and the COD value was reduced. The new method provides an effective and alternative way for the efficient and environmentally friendly extraction of high-quality natural fibers.

## Figures and Tables

**Figure 1 polymers-14-00380-f001:**
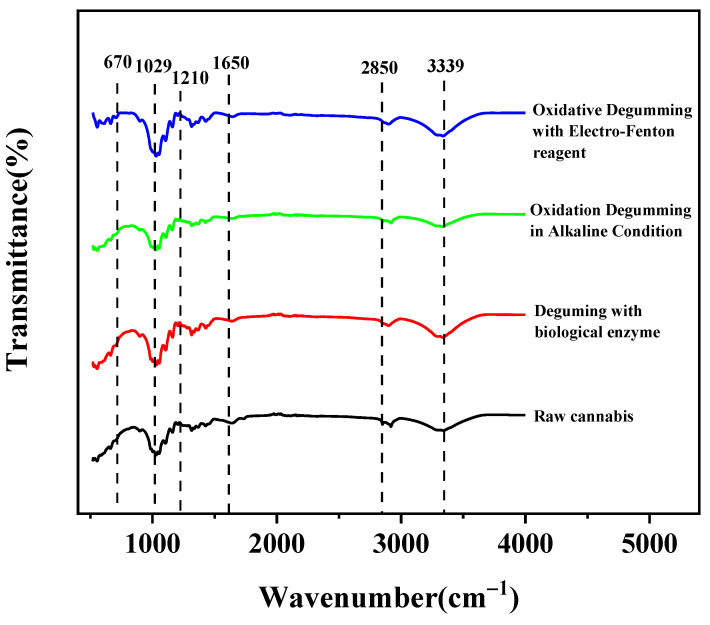
Comparisons of FT-IR spectra of raw cannabis and degummed fibers.

**Figure 2 polymers-14-00380-f002:**
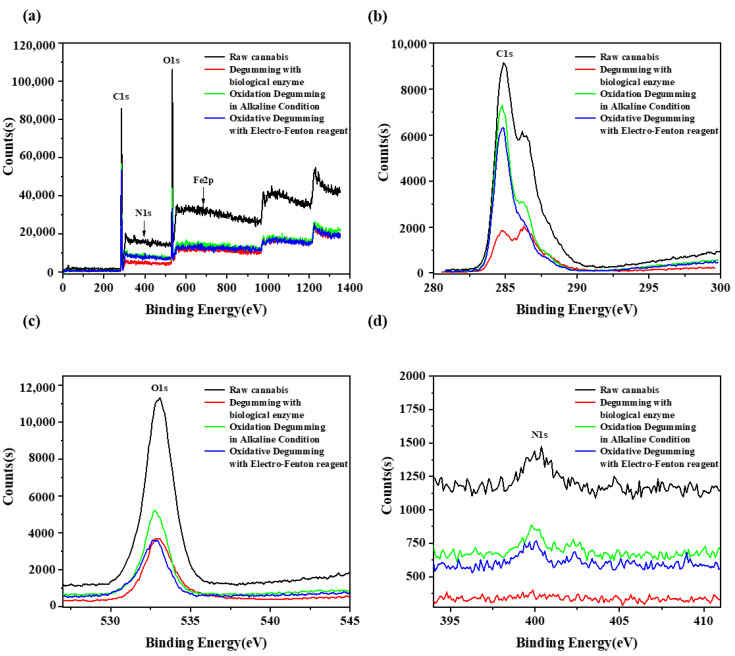
Comparison of XPS spectra (**a**), C1s spectrum (**b**), O1s spectrum (**c**), and N1s spectrum (**d**) of raw cannabis and degummed fiber.

**Figure 3 polymers-14-00380-f003:**
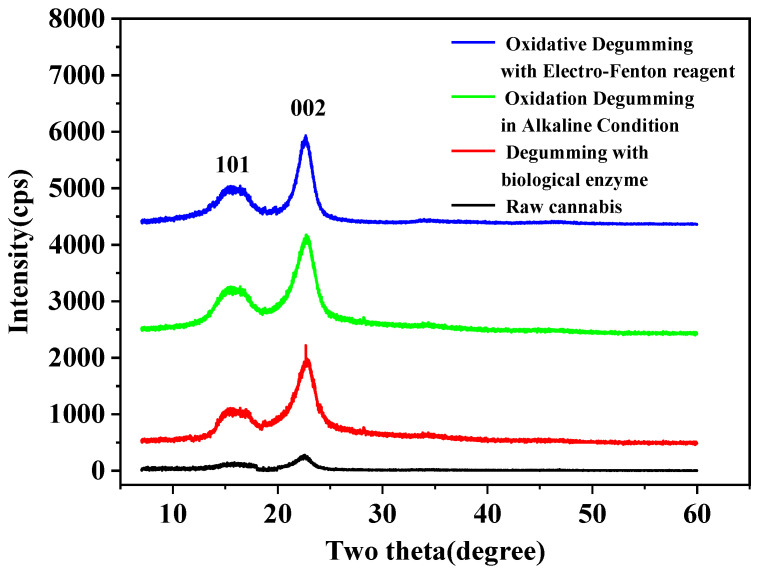
Comparisons of XRD patterns of raw cannabis and degummed fibers.

**Figure 4 polymers-14-00380-f004:**
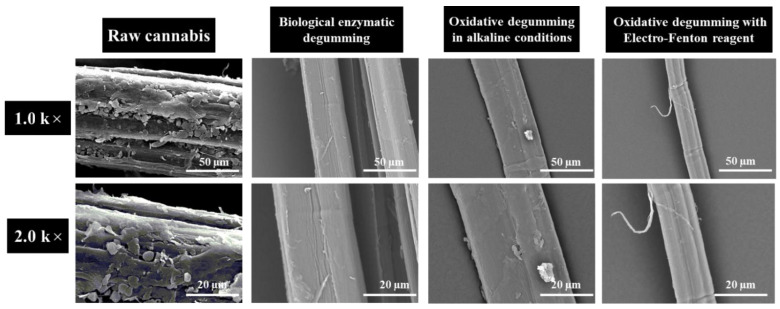
Comparison of the SEM micrographs of raw cannabis and degummed fiber (1.0 k× and 2.0 k×).

**Figure 5 polymers-14-00380-f005:**
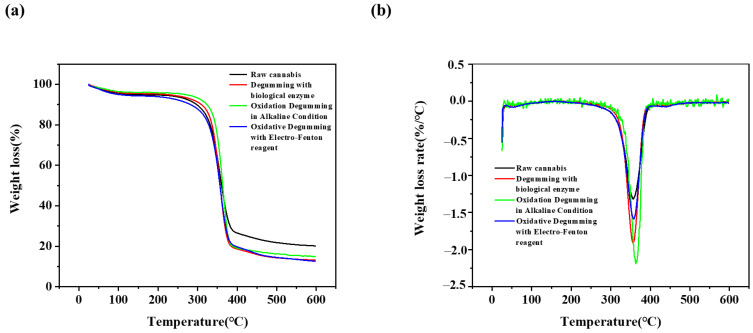
Comparison of thermogravimetric curves of raw cannabis and degummed fibers: (**a**) thermogravimetric curve and (**b**) derivative thermogravimetric curve.

**Figure 6 polymers-14-00380-f006:**
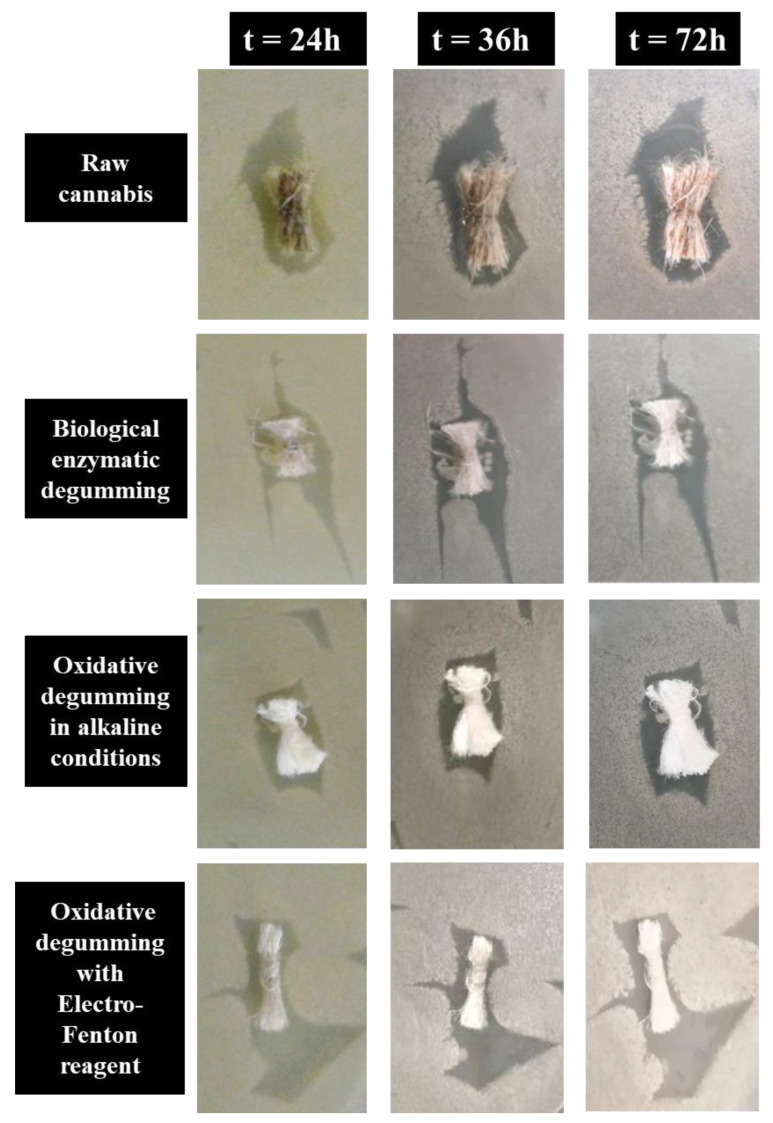
Comparison of antibacterial graphs of raw cannabis and degummed fiber.

**Figure 7 polymers-14-00380-f007:**
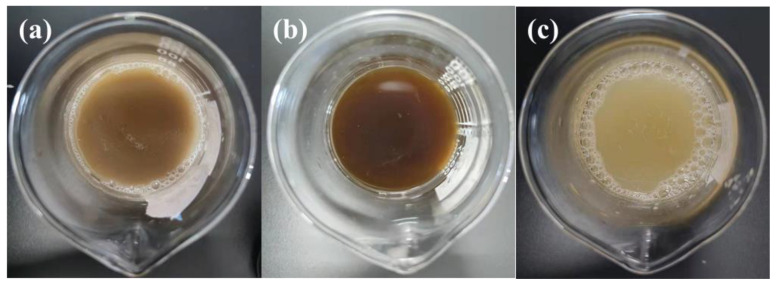
Top view of degumming waste liquid in different processes: (**a**) biological enzymatic degumming, (**b**) oxidative degumming in alkaline conditions, and (**c**) oxidative degumming with electro-Fenton reagent.

**Figure 8 polymers-14-00380-f008:**
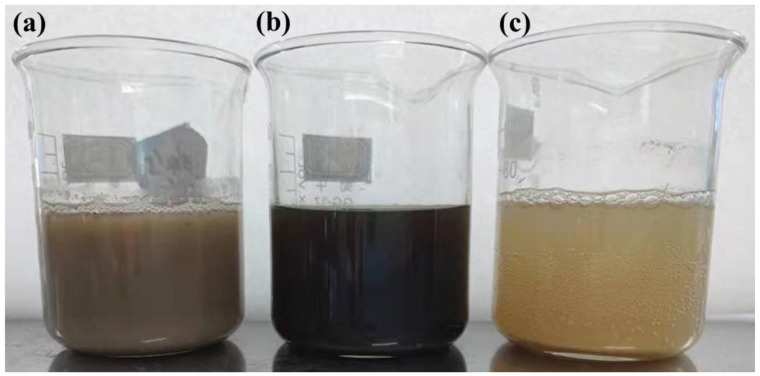
Front view of degumming waste liquid in different processes: (**a**) biological enzymatic degumming, (**b**) oxidative degumming in alkaline conditions, and (**c**) oxidative degumming with electro-Fenton reagent.

**Figure 9 polymers-14-00380-f009:**
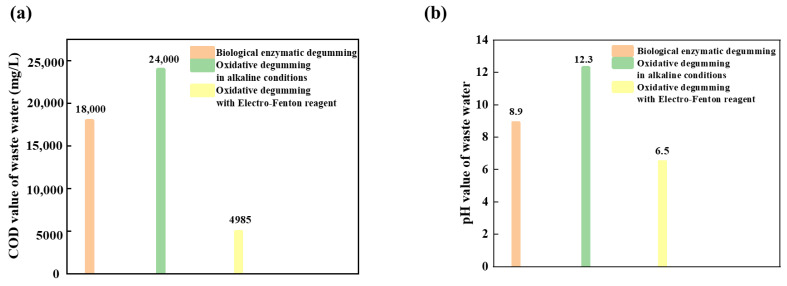
Comparisons of degumming wastewater under different methods: (**a**) COD value of waste water, (**b**) pH value of waste water.

**Table 1 polymers-14-00380-t001:** Chemical composition in cannabis fiber.

Ingredient	Cellulose	Hemicellulose	Lignin	Pectin	Wax	Ash	Water Solubles
Content (%)	49.63	19.32	18.98	6.79	1.32	1.63	2.33

**Table 2 polymers-14-00380-t002:** Conditions of biological enzyme degumming.

	Immersing in Acid Solution	Biological Enzyme Treatment	Oxidation Degumming in Alkaline Condition
Chemical dosage (%)	H_2_SO_4_:1	Laccase:Xylanase:Hemicellulase = 29.0:16.6:54.4	NaOH:5 H_2_O_2_:10 Sulfite:1
Bath ratio	1:20	1:20	1:20
pH		5.0	
Temperature (°C)	Room temperature	50	80
Treatment time (min)	1440	50	60

**Table 3 polymers-14-00380-t003:** Oxidation degumming in alkaline condition.

	First Oxidation	Second Oxidation	Oxidation Degumming in Alkaline Condition
Chemical dosage (%)	H_2_O_2_:1 NaOH:2	H_2_O_2_:1 NaOH:1	NaOH:1
Bath ratio	1:30	1:30	1:30
Temperature (°C)	80	90	50
Treatment time (min)	10	25	40

**Table 4 polymers-14-00380-t004:** Oxidative degumming with electro-Fenton reagent.

	Immersing in Acid Solution	Oxidation	Oxidation Degumming in Alkaline Condition
Chemical dosage (%)	NaOH:1	FeSO_4_∙7H_2_O:5 H_2_O_2_:5 Tripolyphosphate (3-PP):1	NaOH:5 H_2_O_2_:10 Sulfite:1
Bath ratio	1:10	1:10	1:10
pH		6.0	
Temperature (°C)	60	80	80
Treatment time (min)	10	60	40
Voltage(V)		15	
Cathode electrode		Ni-F	

**Table 5 polymers-14-00380-t005:** Chemical composition in cannabis fiber under different degumming methods.

	Cellulose (%)	Lignin (%)	Hemicellulose (%)	Pectin (%)	Wax (%)	Water Solubles (%)	Residual Glue Rate (%)
Raw cannabis	49.63	18.98	19.32	6.79	1.21	4.07	—
Biological enzyme degumming	77.98	9.43	6.84	2.99	0.75	2.01	9.43
Oxidation degumming in alkaline condition	75.5	10.44	11.25	1.62	0.22	0.97	8.81
Oxidative degumming with electro-Fenton reagent	84.57	6.54	4.96	1.43	0.58	1.92	4.77

**Table 6 polymers-14-00380-t006:** Mechanical and physical properties test.

	Tenacity (cN)	Elongation (%)	Diameter (µm)	Whiteness (%)
Raw cannabis	100.03 ± 2.2	0.77 ± 0.5	120.457 ± 2.6	17.32 ± 0.6
Biological enzyme degumming	26.58 ± 0.3	5.03 ± 0.7	64.459 ± 1.7	33.50 ± 0.4
Oxidation degumming in alkaline condition	18.98 ± 0.2	1.08 ± 0.6	13.329 ± 1.5	53.85 ± 0.7
Degumming with electro-Fenton reagent	41.79 ± 0.3	2.32 ± 0.5	15.432 ± 1.3	25.70 ± 0.3

**Table 7 polymers-14-00380-t007:** Binding energy and atomic content of raw cannabis and degummed fiber.

	C1s	O1s	N1s	Fe2p
Raw cannabis	73.35	24.16	0.61	1.89
Biological enzyme degumming	68.59	31.41	—	—
Oxidation degumming in alkaline condition	80.93	18.29	0.78	—
Degumming with electro-Fenton reagent	81.08	16.12	2.8	—

**Table 8 polymers-14-00380-t008:** Comparison of antibacterial bacterial inhibition zone of raw cannabis and degummed fiber.

	ZOI (Diameter, mm)		
	24 h	36 h	72 h
Raw cannabis	9.4 ± 0.3	8.6 ± 0.4	7.4 ± 0.3
Biological enzyme degumming	10.2 ± 0.8	10.1 ± 0.7	9.3 ± 0.7
Oxidative degumming under alkaline conditions	9.2 ± 0.6	8.7 ± 0.5	8.4 ± 0.5
Oxidative degumming of electro-Fenton reagent	10.4 ± 0.5	9.8 ± 0.6	9.5 ± 0.7

## Data Availability

Not applicable.

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
