# Peer review of "Separation and Characterization of Cellulose Fibers from Cannabis Bast Using Foamed Nickel by Cathodic Electro-Fenton Oxidation Strategy"

_polymers, 2022, doi:10.3390/polym14030380_

Round 1

Reviewer 1 Report

In the paper entitled Separation and Characterization of Cellulose Fibers from Cannabis Bast Using Foamed Nickel as Cathodic Electro-Fenton Oxidation Treatment following modifications are suggested to be made before it can be published

1: The introduction is written in thesis form. The authors should indicate their motivation and novelty of  work.

2: In Figure 4 the authors have mentioned differences from Fig4 b to d. However, these differences are not so clear to me. Moreover, hoe the authors conclude about gumming /degumming from SEM images. It is interesting to know because apparently one can not see this difference.

3: The authors called photos to TGA plots in Figure 5. First these are not photos but TGA gives you plots or graphs. Secondly the images are blurred. lease provide clear one.

Reviewer 2 Report

Review Report

The work entitled “Separation and Characterization of Cellulose Fibers from Cannabis Bast Using Foamed Nickel as Cathodic Electro-Fenton Oxidation Treatment” focuses the separation and characterization of bast fibers with electro-Fenton (EF) system enabled with a nickel-foam (Ni-F) cathode in weak acid to degum cannabis fiber and reducing the content of pollutants in degumming  wastewater. The authors work is appreciable; however, some of the aspects of this work can be improved.

This manuscript could be acceptable for publication after its improvement.

Comments and Suggestions

  1. In the `Introduction` section, specifically Cathodic Electro-Fenton Oxidation Treatment in reducing the content of pollutants should be mentioned more clearly.
  2. In “antibacterial test’ under the `Results and discussion` section, kindly mention the standard error value of the mean for bacterial inhibition zone for different times for better understanding.
  3. In “XRD analysis” under the `Results and discussion` section, the 2 theta intensity of raw cannabis fibers seems less compared to the others. Kindly provide the crystalline size comparison for all the fibers to understand whether they have any significant different in regards to their crystallinity.
  4. In Table.6 in “Mechanical and Physical Test” under the `Results and discussion` section, kindly provide the standard error of the mean value of the
  5. In, ”Chemical composition of different degumming methods of cannabis fiber” under the `Results and discussion` it is mentioned that ‘. The lignin content decreased from 18.98% to 202 9.43%, and the hemicellulose content decreased from 19.32% to 6.84%. Therefore, the biological enzyme method can effectively remove lignin and hemicellulose’ Why lignin and hemicellulose removal is significant for the ” reducing the content of pollutants” with degummed fibers. What will be the specific effect of lignin and hemicellulose when they are not removed. Please explain.

-------------------------------------------------***----------------------------------------------------

Round 2

Reviewer 1 Report

Caption of Figure 5 needs to be revised. Why are the authors not able to differentiate between spectra, diagrams and plots? What does TG and DTG means on the Y-axis of Figure 5? Please check literature for correction.

The text on all figures should be readable.
